# Optimized Convolutional Neural Network Recognition for Athletes’ Pneumonia Image Based on Attention Mechanism

**DOI:** 10.3390/e24101434

**Published:** 2022-10-08

**Authors:** Hui Zhang, Ruipu Ma, Yingao Zhao, Qianqian Zhang, Quandang Sun, Yuanyuan Ma

**Affiliations:** 1College of Physical, Henan Normal University, Xinxiang 453007, China; 2College of Software, Henan Normal University, Xinxiang 453007, China; 3College of Computer and Information Engineering, Henan Normal University, Xinxiang 453007, China

**Keywords:** image recognition, image enhancement, lung image, athlete, pneumonia

## Abstract

After high-intensity exercise, athletes have a greatly increased possibility of pneumonia infection due to the immune function of athletes decreasing. Diseases caused by pulmonary bacterial or viral infections can have serious consequences on the health of athletes in a short period of time, and can even lead to their early retirement. Therefore, early diagnosis is the key to athletes’ early recovery from pneumonia. Existing identification methods rely too much on professional medical knowledge, which leads to inefficient diagnosis due to the shortage of medical staff. To solve this problem, this paper presents an optimized convolutional neural network recognition method based on an attention mechanism after image enhancement. For the collected images of athlete pneumonia, we first use contrast boost to adjust the coefficient distribution. Then, the edge coefficient is extracted and enhanced to highlight the edge information, and enhanced images of the athlete lungs are obtained by using the inverse curvelet transformation. Finally, an optimized convolutional neural network with an attention mechanism is used to identify the athlete lung images. A series of experimental results show that, compared with the typical image recognition methods based on DecisionTree and RandomForest, the proposed method has higher recognition accuracy for lung images.

## 1. Introduction

Pneumonia is one of the most common infectious diseases in clinics. Because of its short onset and complex causes, the early identification of pneumonia plays an important role in the diagnosis and treatment of pneumonia. X-ray examination has been widely used in image examination because of its advantages such as low radiation, low cost and high practicability. However, X-ray chest radiography also has some deficiencies [1], such as low image resolution and overlapping and intersecting organs in the image, which increases the difficulty of identification and easily leads to misdiagnosis and misdiagnosis by doctors. In addition, lung health is particularly important for athletes. If pneumonia is not detected and treated in time, athletes will be unable to participate in vigorous exercise due to breathing difficulties, which delays the resumption of training and impacts the golden period of athletes, thus shortening their sports life. Therefore, the computer-aided diagnosis of athlete pneumonia [2] has become one of the important topics in the field of sports medicine in recent years.

In order to quickly and accurately identify athlete pneumonia, researchers have invested significant energy and put forward many ideas. In summary, the current research works can be divided into three categories: the first is the pneumonia image recognition method based on DecisionTree. Wendy Webber Chapman et al. applied the DecisionTree algorithm to pneumonia image recognition. They used Quinlan’s SEE5.0 DecisionTree Software (RuleQuest ResearchPty Ltd., Empire Bay, New South Wales, 2257, Australia) to learn DecisionTree from data and use the DecisionTree algorithm to identify pneumonia. The accuracy of this method for pneumonia image recognition is 78% [3]. The second is a pneumonia recognition method based on RandomForest. Akgundogdu Abdurrahim proposed a RandomForest algorithm to classify chest X-ray pneumonia images. This method first extracts features from an image using a two-dimensional discrete wavelet transform. The features obtained by the wavelet method are marked as normal and pneumonia, respectively, and applied to the classifier for classification. The results also showed that only 130 of 4234 pneumonia cases were misdiagnosed [4]. The third is the deep learning algorithm represented by convolutional neural networks. Jakhar K et al. used DCNN, a large data prediction model optimized for the RandomForest algorithm, to identify pneumonia images with an accuracy of 84% [5]. Pranav Rajpurkar et al also proposed a chest X-ray pneumonia recognition algorithm (ChexNet), which uses the ChestX-ray 14 dataset published by the NIH (National Institutes of Health) as training test data for in-depth learning to identify 14 lung diseases, with an accuracy of 76.8% [6]. El Asnaoui Khalid and Youness Chawki used DensNet201 to identify pneumonia with an accuracy of 88.09% [7]. All the above methods can effectively identify pneumonia, which is much more efficient than traditional methods, but there is still a problem of low recognition accuracy.

To solve the problem, this paper presents an optimized convolutional neural networks based on an attention mechanism for lung image recognition after image enhancement. Firstly, for the low frequency coefficient matrix after curvelet transformation, an adaptive contrast boost method is proposed, which makes the coefficient distribution in the matrix more uniform, enhances the contrast of the image, improves the value of the image and is easier to process by the machine. Second, the edge coefficient method is used to enhance the extracted edge coefficient to improve the image contrast and enhance the edge factor. Then, the enhanced image is obtained by inverse curvelet transformation, which is more conducive to image edge recognition. Finally, an optimized convolutional neural network incorporating an attention mechanism is used to identify the images. It was found that the enhanced image has a higher recognition rate and can be effectively used in the field of image recognition of athlete lungs to improve the accuracy of recognition of athlete pneumonia.

The rest of this paper is arranged as follows. Section 2 describes the related processes and principles of image enhancement. Section 3 gives the process of optimizing convolutional neural networks based on an attention mechanism. Section 4 describes the process of the convolutional neural network identification method proposed in this paper based on an attention mechanism. After that, the experimental results of the optimized convolutional neural networks based on an attention mechanism before and after enhancement and the experimental results of the enhanced optimal convolutional neural networks compared with RandomForest and DecisionTree are given macroscopically in Section 5. Finally, the full text is summarized in Section 6.

## 2. Image Enhancement

This section will optimize image enhancement from three perspectives: contrast, edge coefficient and overall enhancement contrast [8], so as to improve the accuracy of the optimized convolutional neural network based on an attention mechanism in the recognition of athlete pneumonia images.

### 2.1. Processing of Contrast Improvement

The contrast of the image reflects the difficulty of image recognition. That means that the greater the contrast, the easier it is to recognize for people and machines. Thereby, increasing the contrast of the image can reduce the difficulty of image recognition. The image contour and contrast mainly come from the low-frequency coefficient matrix after curvelet transformation. Therefore, the low-frequency coefficient matrix needs to be processed accordingly to achieve the image enhancement effect [9]. Generally, we perform curvelet forward transformation on the image at first to obtain the coefficients cj,dk1,k2. Let the coefficient matrix Ii,j consist of coefficients cj,dk1,k2, with row *i* and column *j*. Here, we use contrast boosting to process the low-frequency coefficient matrix. The specific process is shown in Algorithm 1.
**Algorithm 1:** Contrast improvement**Input:** Coefficient matrix Matrixoriginal;**Output:** Transformed coefficient matrix Matrixprocessed; 1:Normalize the coefficient values in Ii,j to the 0,1 interval; 2:Set step=0.1, divide 0,1 into *n* original intervals, then count the number of coefficients in interval; 3:Calculate the frequency of each interval pi (frequency is the percentage of coefficients in each interval as a percentage of the total number); 4:Use pi to calculate the new interval corresponding to each interval after the transformation, *n* intervals are expressed as:0,p1,p1,p1+p2,…,p1+…+pn−1,p1+…+pn; 5:Use Equation (Equation 1) to transform the value of the interval to obtain the transformed value *b*,
(1)b=p1×astep,i=1p1+…+pi−1+pi×a−i−1×stepstep,1<i<np1+…+pi−1+1−p1+…+pi−1×a−i−1×step1−i−1×step,i=n
 6:Perform anti-normalization to obtain the transformed coefficient matrix.


Example 1: $Matrix_{original}=\left[\begin{array}{cccc} 11.3 & 254.2 & 1.0 & 2.3\\ 19.6 & 200.1 & 221.9 & 230.6\\ 5.8 & 9.7 & 6.1 & 210.9\\ 219.3 & 228.2 & 8.8 & 236.8 \end{array}\right]$; the coefficient matrix after the above contrast enhancement is $Matrix_{processed}=\left[\begin{array}{cccc} 35.3 & 254.2 & 1.0 & 5.3\\ 63.0 & 146.7 & 192.2 & 209.9\\ 17.0 & 30.0 & 18.0 & 169.3\\ 186.8 & 205.3 & 27.0 & 221.6 \end{array}\right]$.

Comparing the two matrices, it can be seen that the coefficients in $Matrix_{processed}$ are more uniformly distributed than those in matrix $Matrix_{original}$.

It can be concluded from the above that after Algorithm 1, the coefficient matrix achieves the effect of improving image contrast, and finally achieves the goal of easy image recognition.

### 2.2. Processing of Edge Coefficient Extraction and Enhancement

In order to recognize the athlete lung images more accurately, this paper uses the edge coefficient extraction and enhancement operation to further enhance the image. Curvelet transformation is a multifaceted transformation, which includes scale, direction and resolution. After the curvelet transform, edge coefficients are enhanced, which can achieve the purpose of enhancing the image. The enhancement equation of the edge coefficient is as follows,
(2)$$c_{j,d}^{\prime }\left(k_{1},k_{2}\right)=\left\{\begin{array}{c} M-\frac{{\left(M-c_{j,d}\left(k_{1},k_{2}\right)\right)}^{2}}{M-m},M_{jdk}>M_{edge}\\ c_{j,d}\left(k_{1},k_{2}\right),else \end{array}\right.\;$$
where, $c_{j,d}\left(k_{1},k_{2}\right)$ represents the curvelet coefficient on scale, direction is *d*, and position $\left(k_{1},k_{2}\right)$, $c_{j,d}^{\prime }\left(k_{1},k_{2}\right)$ represents the processed curvelet coefficient. $M_{edge}$ represents the sum of the absolute values of the set of mapping coefficients, *M* represents the maximum coefficient satisfying condition $M_{dk}>M_{edge}$ and *m* represents the minimum coefficient satisfying condition $M_{dk}>M_{edge}$.

The specific steps of edge coefficient extraction and enhancement are shown in Algorithm 2.
**Algorithm 2:** Edge coefficient extraction and enhancement**Input:** Edge coefficient extraction and enhancement;**Output:** Enhanced image after inverse transformation;
 1:Based on the mapping characteristics of curvelet coefficients, the mapping coefficient matrices $Matrix_{mapping}$ of the original coefficient matrix in different scales and directions are obtained; 2:Calculate the sum of the absolute values of the coefficients within the matrix; 3:Set step=0.1 and divide $\left[0,1\right]$ into *n* original intervals; then, count the number of coefficients in the interval; 4:Set a threshold value τ; the coefficient whose sum of absolute values is greater than the τ is considered as the edge coefficient; 5:Perform inverse curvelet transform on α to obtain the enhanced image.


Through Algorithm 2, the athlete pneumonia image is further enhanced.

## 3. Optimized Convolutional Neural Network Based on Attention Mechanism

Convolutional neural networks can distinguish medical images and the video data. Using the optimized convolution neural network with an attention mechanism, the enhanced image is recognized. According to the different importance of features, different weights are given to features on different channels to highlight important features and suppress secondary features [10], so as to improve the accuracy of recognition.

### 3.1. Processing of Convolutional Neural Network

The hierarchy of a convolutional neural network includes the convolutional layer, excitation layer, pooling layer and full connection layer, which are shown in Figure 1. Among them, the convolution layer is one of the methods to extract image features by convolution neural networks. It senses local image features through a local perception field and reduces the computational load on the network by weight sharing [11].

In the convolution layer, data exists in 3D. For example, if a region in a gray image is in the input layer, one feature will be obtained by computing. Similarly, a region in a color image will have three features. The features of the previous layer are convoluted with the corresponding convolution kernel, which outputs new features. The convolutional layer equation is as follows,
(3)$$z_{u,v}^{\left(l\right)}=\sum_{i=-\infty }^{\infty }\sum_{j=-\infty }^{\infty }x_{i+u,j+v}^{\left(l-1\right)}\cdot {k_{rot}}_{i,j}^{\left(l\right)}\cdot x\left(i,j\right)+b^{\left(l\right)}$$
(4)$$x\left(i,j\right)=\left\{\begin{array}{c} 1,0\leq i,j\leq n\\ 0,others \end{array}\right.$$
(5)$$a_{u,v}^{\left(l\right)}=f\left(z_{u,v}^{\left(l\right)}\right)$$
where l−1 is the input layer, its input feature map is $X^{\left(l-1\right)}\left(m\times m\right)$ and the convolutional kernel corresponding to the feature is $k^{\left(l\right)}\left(n\times n\right)$; then, a bias unit $B^{\left(l\right)}$, is added to each output, resulting in the output of the convolutional layer being $Z^{\left(l\right)}\left(\left(m-n+1\right)\times \left(m-n+1\right)\right)$.

Convolutional kernels are also known as filters; each convolutional kernel has three dimensions: length, width and depth. In each convolutional layer, the length and width of the convolution kernels are specified artificially. Because the depth of the convolutional kernel is the same as the current image (the number of feather maps), it is not necessary to specify. This paper decides to use 3×3 convolutional kernel.

In order to improve the expression ability of linear models, an activation function is added after the convolution layer when the convolutional neural network is built. The function of LeakyReLU (Leaky Rectified Linear Unit) has the advantages of no gradient dissipation, fast convergence, increasing network sparsity and less computation. Therefore, the LeakyReLU function is selected as the activation function in this paper. The analytic equation of the LeakyReLU function is as follows,
(6)$$LeakyReLU\left(x\right)=\left\{\begin{array}{c} x,x>0\\ ax,x\leq 0 \end{array}\right.$$

The purpose of the pooling layer is to reduce the characteristic graph and improve the robustness of the model. Among many pooling operations, maximum pooling is the most widely used and has the best effect. The maximum pooling is as follows,
(7)$$h_{m,j}=\max_{i\in N_{m}}a_{i,j},forj=1,\ldots \;K$$
(8)$${z_{i,j}}^{\left(l+1\right)}=\beta^{\left(l+1\right)}\sum_{u=ir}^{(i+1)r}\sum_{v=jr}^{1(j+1)r}h_{u,v}^{\left(l\right)}+b^{\left(l+1\right)}$$
(9)$$a_{i,j}^{\left(l+1\right)}=f\left(z_{i,j}^{\left(l+1\right)}\right)$$

Thus, in this paper, maximum pooling is adopted as a pooling operation in the convolutional neural network. Here, the weight of each cell used is $\beta^{\left(l+1\right)}$. Each convolution operation is thrown with a bias element $b_{ij}^{\left(l+1\right)}$, then giving the pooled output as a ${a_{ij}}^{\left(l+1\right)}\left(\frac{m-n+1}{r}\times \frac{m-n+1}{r}\right)$ order matrix. After the maximum pooling, the image will retain more texture edge information. After enhanced the edge of the athlete lung image, these pneumonia feature information will be more retained in the convolutional neural network. This is more conducive to enhancing the athlete’s lung image to improve the accuracy of athlete pneumonia image recognition.

There are some inherent problems with the recognition process in the traditional convolutional neural network that will reduce the recognition accuracy of the convolutional neural network, such as high learning rate, overfitting and large oscillation at the optimal point. Therefore, it is necessary to optimize the convolutional neural network.

### 3.2. Optimization of Convolutional Neural Networks

The exponential decay learning rate strategy, dropout method, and Adam algorithm can all optimize the convolutional neural network model to improve its generalization ability and prevent overfitting.

#### 3.2.1. Exponential Decay Learning Rate Strategy

Generally, the method of gradient descent is used to adjust the learning rate of the model to ensure that the machine learning model finds the optimal solution [12] in the learning process. If the learning rate is too large, the model will miss the optimal solution. Otherwise, it will fail to reach the optimal solution. Therefore, we need to constantly adjust the learning rate during the model training process.

In this paper, the exponential decay learning rate strategy is used to update the model learning rate. When the learning rate is large, the network parameters update quickly and the local convergence is fast at the initial stage of model training. As the number of iterations increases, the model will reverberate to the global optimal point. At this time, a smaller learning rate can make the model converge. Learning rate is a super parameter that guides us as to how to adjust the weight of the network through the gradient of the loss function. By setting the learning rate, the speed of parameter updating is controlled. The equation is as follows,
(10)decayed_learning=learning_rate×decay_rate∧(global_step/decay_steps)
where *decayed_learning* is the exponential decay learning rate, *learning_rate* is the initial learning rate, *decay_rate* is the decay coefficient, *global_step* is the round iteration number, *decay_steps* is the control decay rate.

#### 3.2.2. Dropout Method

In the process of machine learning, if the model parameters are too many and the number of samples used for training is too small, overfitting of the trained model will occur. Adding dropout to network training can effectively mitigate the occurrence of overfitting [13].

Dropout is a regularization method. In training, the hidden layer of neurons are suppressed with probability P and part of the hidden layer neurons are discarded, while the parameter values of the discarded nodes are retained. Only the parameter values of the activated neurons are updated during error backpropagation. The process will be repeated in the next training and each training will obtain a different neural network, which will finally be integrated.

Dropout can learn the corresponding features in different subsets of hidden layers to reduce the dependence between features and prevent overfitting. The standard neural network and the neural network diagram with dropout applied are shown in Figure 2.

#### 3.2.3. Adam Algorithm

In machine learning, the Adam algorithm is a method to optimize the loss function with a one-step degree [14]. The algorithm has high accuracy and low storage overhead, and can greatly avoid model shaking at its best. Therefore, it is suitable for neural networks with large-scale datasets and parameters. The Adam algorithm flowchart is shown in Figure 3.

where α is stepsize, $\beta_{1}$ and $\beta_{2}$ are exponential decay rates for the moment estimates, $m_{0}$ is first moment vector after initialization, $v_{0}$ is second moment vector after initialization, *t* is timestep after initialization, $\theta_{0}$ is parameter vector after initialization, $f\left(\theta \right)$ is the stochastic objective function with parameters θ, $g_{t}$ is gradients w.r.t. the stochastic objective at timestep *t*, $m_{t}$ is biased first moment estimate after the update, $v_{t}$ is biased second raw moment estimate after the update, ${\hat{m}}_{t}$ is bias-corrected first moment estimate, ${\hat{v}}_{t}$ is bias-corrected second raw moment estimate and $\theta_{t}$ is parameters after the update.

### 3.3. Attention Mechanism

In order to make the optimized convolutional neural network ignore the irrelevant information and pay more attention to the key information, this paper adds an attention mechanism module to the optimized convolutional neural network. It is an attentional module called Convolutional Block Attention Module (CBAM) that combines spatial and channel dimensions.

CBAM is an end-to-end, simple and effective computer visual attention module. In a convolutional neural network, CBAM can input any intermediate feature map along two independent dimensions: spatial and channel [15]. Then, the attention is multiplied by the input feature map to perform adaptive feature refinement on the input feature map [16]. Finally, it is seamlessly integrated into the framework of any convolutional neural network.

The calculation method of the channel attention model in CBAM is shown in Equation (Equation 11),
(11)$$M_{c}\left(F\right)=\sigma \left(MLP\left(AvgPool\left(F\right)\right)+MLP\left(MaxPool\left(F\right)\right)\right)\;\;$$
where *MLP* is a multi-layer perceptron and σ is a sigmoid function, *AvgPool* is the averaging pooling operation, *MaxPool* is the maximum pooling operation and *F* is the feature map of the input.

In the spatial attention process of CBAM, first, perform the maximum pooling and average pooling on the input characteristic map. Next, a convolution is used to reduce the dimensions of the feature maps to one dimension along the channel direction. The process can be described as in Equation (Equation 12),
(12)$$M_{s}\left(F\right)=\sigma \left(f^{n\times n}\left(\left[AvgPool\left(F\right);MaxPool\left(F\right)\right]\right)\right)\;\;$$
where $f^{n\times n}$ is the convolution of n×n magnitude.

The overall attention process of CBAM can be described as in Equation (Equation 13),
(13)$$\begin{array}{c} F^{\prime }=M_{c}\left(F\right)⊗F\\ F^{\prime \prime }=M_{s}\left(F^{\prime }\right)⊗F^{\prime } \end{array}\;\;$$
where $F^{\prime \prime }$ is the final output feature map and ⊗ is the multiplication operation.

After several experiments, this paper adds a CBAM attention mechanism module after each excitation layer of the optimized convolutional neural network. The specific adding method is shown in Figure 4.

## 4. Optimized Convolutional Neural Network Recognition Method for Athletes’ Pneumonia Image Based on Image Enhancement and Attention Mechanism

Due to the limitation of medical image imaging conditions, the contrast of the image is low. Details are often high-frequency signals, but they are often embedded in a large number of low-frequency background signals, which reduces the visibility of the image [17]. Therefore, appropriately improving the contrast of the image can improve the accuracy of image recognition.

Image edge is one of the most basic characteristics of an image, which often carries most of the information. The edge exists on the irregular and unstable structure of the image, that is, at the mutation point of the signal. These points give the position of the image contour, which often contains important features required for image edge detection. Therefore, it is necessary to extract and enhance the edge coefficient of the image to further improve the accuracy of image recognition.

For enhanced athlete lung images, this paper adopts an optimized convolutional neural network based on an attention mechanism to recognize the image.

The optimized convolutional neural network based on image enhancement and an attention mechanism (CIEA) recognizes the athlete pneumonia image in three steps: image acquisition, image enhancement and image recognition. First, the images of the athlete lungs are collected. Then, contrast enhancement and edge coefficient enhancement are performed on the collected images of athlete lungs to improve the image recognition accuracy. Finally, the optimized convolutional neural network based on an attention mechanism can recognize the optimized lung images of athletes, which improves the efficiency of diagnosis and treatment to help athletes to resume training as soon as possible. The image enhancement parts uses the method of contrast enhancement and edge enhancement of the second part of this paper. In the image recognition part, the optimized convolutional neural network based on an attention mechanism in Part 3 of this paper is adopted.

The specific steps of the CIEA method to identify athlete pneumonia are shown in Algorithm 3.
**Algorithm 3:** CIEA method for identifying athlete pneumonia**Input:** Preprocessed gray images;**Output:** P;
 1:Input the preprocessed gray images; 2:Calculate the sum of the absolute values of the coefficients within the matrices; 3:Carry out curvelet forward transformation on the images to obtain the transformed coefficient matrices $Matrix_{original_{1...n}}$; 4:$Matrix_{original_{1...n}}$ are processed by contrast enhancement to obtain the coefficient matrices $Matrix_{processed_{1...n}}$ according to Algorithm 1; 5:Based on the mapping characteristics of curvelet coefficients, the mapping coefficient matrices $Matrix_{mapping_{1...n}}$ of coefficient matrices $Matrix_{processed_{1...n}}$ in different scales and directions are obtained; 6:The edge coefficients are judged according to Algorithm 2; then, the enhanced edge coefficient matrices are calculated according to Equation (Equation 2). Finally, the curvelet inverse transformation is performed on the matrices and the images with contrast enhancement and edge coefficient enhancement are obtained; 7:Put the enhanced images into the trained network model; 8:Return the accuracy of identifying pneumonia P.


The overall framework of the CIEA method to identify athlete pneumonia is shown in Figure 5 below.

## 5. Experimental Results

In order to verify the high recognition accuracy of CIEA method proposed to this paper for athlete pneumonia, a series of experiments were carried out and the results were compared with the DecisionTree image recognition algorithm and the RandomForest image recognition algorithm.

### 5.1. Data Preprocessing

To improve the authenticity of the research, this paper selected a large number of athletes from professional sports colleges. X-ray images of the above athletes’ lungs were collected. There were 2806 men and 2806 women, totaling 5612, accounting for 30% of the unit athletes in 2020. The ages were between 18 and 25 years old. The shortest training period was only 1 year and the longest training period was 5 years. Figure 6 and Figure 7 are the medical images of the part collected.

A total of 5612 original grayscale images were collected and randomly clipped to 224×224 grayscale images, with the pixel value divided by 255 to normalize the data, thereby accelerating the training process. Finally, the data were divided into a training set, a validation set and a test set at a 8:1:1 ratio, which is convenient for the training and evaluation of subsequent models.

### 5.2. CIEA Method Recognition Experiment

A series of experiments were carried out to verify the high accuracy of the proposed method in lung image recognition of athletes.

This paper uses the TensorFlow framework in the experiment, using several models and parameters. After many experimental comparisons, an eleven-layer convolutional neural network is finally adopted. The network model built in this paper has a simple structure, which consists of five layers of convolution, five layers of pooling and one output fully connected layer. The whole network structure uses LeakyReLU instead of ReLU as the activation function, which can effectively solve the Dead ReLU problem. Binary cross entropy is used as the loss function of the network; the formula of the loss function is as follows:(14)$$Loss=-1/N\sum_{i=1}^{N}y_{i}\cdot \log\left(p\left(y_{i}\right)\right)+\left(1-y_{i}\right)\cdot \log\left(1-p\left(y_{i}\right)\right)$$

At the same time, the network adds the dropout operation after layer 2, layer 4 and layer 5 convolution layers and adds the normalized layer after each convolution layer (batch normalization, BN). An attention mechanism module is added in each down sampling layer.

Adam is used as the optimizer, the batch size is set to 16, the number of training cycles epochs is set to 300 and the initial learning rate is set to $10^{{}^{-4}}$. We adopt exponential decay learning rate strategy, dropout and the Adam algorithm to improve model performance. The exponential decay learning rate strategy ensures that the model parameters update faster and converge at the global optimal point. Dropout reduces dependence between features to prevent overfitting. The Adam algorithm corrects the gradient mean and the gradient square mean using the number of iterations and the exponential decay rate to make the prediction of the gradient more accurate.

In order to test the effectiveness of image enhancement methods, this paper uses the methods of contrast enhancement and edge coefficient enhancement to enhance the athlete lung medical images. Five medical images were taken as examples (1, 2, 3 without pneumonia; 4, 5 with pneumonia) and image enhancement was performed on these five images. The original images, the images with enhanced contrast, the images with enhanced edge information, and the images with integrated enhancement are shown in Figure 8.

The comparison results of the optimized convolutional neural network recognition algorithm based on an attention mechanism before and after image enhancement are shown in Table 1 below.

As can be seen from Table 1, in the training set, the accuracies of the model trained with original data and enhanced data are very high, with little difference between the two. However, on the validation set, the model trained with the enhanced data is significantly more accurate than with the original data, improving by 4%; on the test set, the improvement is 2%. The experimental results show that enhancing the image can help improve the generalization ability of the model and the accuracy of image recognition.

The optimized convolutional neural network with an attention mechanism is used trained on the datasets before and after enhancement; the results are shown in Figure 9, Figure 10, Figure 11 and Figure 12.

In Figure 9 and Figure 10, the blue curve is the accuracy of training the model with the original data, and the orange curve is the accuracy of training the model with the enhanced data. It can be seen from the two figures that, with the increase of the number of training rounds, the enhanced data have higher accuracy than the original data in the neural network model. Moreover, in the validation set, the variation range of original data accuracy is very unstable, which indicates that enhancing the data is helpful to improve the generalization ability of the model.

In Figure 11 and Figure 12, the blue curve is the loss value of the training model with the original data, and the orange curve is the loss value of the training model with the enhanced data. As can be seen from the two graphs, the model trained with enhanced data can converge better and faster, and the change of the model loss values trained with original data on the validation set are very unstable, which further indicates that image enhancement is conducive to improving the representation ability of the model and, thus, improving the accuracy of the model.

### 5.3. Comparative Experiment of Different Recognition Methods

In order to compare the method proposed in this paper with the traditional DecisionTree pneumonia recognition algorithm in machine learning and the RandomForest pneumonia recognition algorithm, we conducted the following comparative experiments.

#### 5.3.1. Pneumonia Recognition Algorithm Based on DecisionTree

Wendy Webber Chapman et al. applied DecisionTree algorithm to pneumonia image recognition [3]. Specifically, they used Quinlan’s SEE5.0 Decision Tree Software (RuleQuest Research Pty Ltd., Canberra, Australia) to learn decision trees from data. Given the attribute-value (a-v) pair vector for each training case, the algorithm calculates the information gain for each contribution and selects the one with the largest value as the first branch of the tree. The winning a-v pairs are no longer considered and the information gain of the remaining a-v pairs is calculated; then, the pair with the highest score is selected and placed on the next branch of the tree. This process is repeated until most training cases are successfully classified.

#### 5.3.2. Pneumonia Recognition Algorithm Based on RandomForest

Akgundogdu Abdurrahim proposed a RandomForest algorithm based on chest X-ray images to classify the diagnosis of pediatric pneumonia [4]. The algorithm first extracts features from the image using a two-dimensional discrete wavelet transform. The features obtained by the wavelet method are marked as normal and pneumonia, respectively, and applied to the classifier for classification. In addition, 5856 X-ray images are classified using a random forest algorithm. A 10-fold cross-validation method is used to assess the success of the proposed method and to ensure that the system avoids overfitting.

#### 5.3.3. Pneumonia Recognition Algorithm Based on ChexNet

Pranav Rajpurkar et al proposed a chest X-ray pneumonia recognition algorithm (ChexNet), which is a 121-layer convolutional neural network that inputs a chest X-ray image and outputs the probability of pneumonia along with a heatmap localizing the areas of the image most indicative of pneumonia [6].

Based on the athlete lung images, the comparison results using the DecisionTree, Random- Forest, ChexNet and CIEA recognition algorithms are shown in Table 2 below.

From Table 2, we can see that the accuracy of the optimized convolutional neural network based on an attention mechanism is higher than that of traditional machine learning and traditional deep learning for both the validation and test set. On the validation set, it is 9.76% higher than the DecisionTree algorithm, 9.19% higher than the RandomForest and 13.99% higher than the ChexNet. On the test set, it is 11.96% higher than the traditional machine learning algorithm. Because DecisionTree and RandomForest require that input data be one-dimensional, the pictures must be compressed into one dimension to train them. This operation loses the spatial characteristics of the images, which limits the accuracy of the models. Convolutional neural networks can directly input two-dimensional pictures as datasets and use the convolution kernel to extract spatial features, which can better identify object features. The result is therefore better. It is 16.41% higher than the traditional deep learning algorithm because CIEA uses the enhanced dataset and adds an attention mechanism module to the neural network. Therefore, the optimized convolutional neural network based on an attention mechanism can recognize the enhanced pneumonia images more accurately.

## 6. Conclusions

In this paper, an optimized convolutional neural network recognition method based on image enhancement and an attention mechanism is proposed to recognize athlete pneumonia. First, we collected medical images of athlete lungs. Then, image enhancement methods, such as enhancing contrast and enhancing edge coefficients, were used to enhance the lung image of athletes. Finally, this paper used the optimized convolutional neural network recognition method based on an attention mechanism to recognize images; the results show that the accuracy of the method proposed in this paper was improved by 11.96% on the test set, indicating that this method can help to improve the generalization ability of the model and make the model more accurate. At the same time, the recognition rate of athlete pneumonia by the optimized convolutional neural network based on an attention mechanism is as high as 97.31%. 

## Figures and Tables

**Figure 1 entropy-24-01434-f001:**
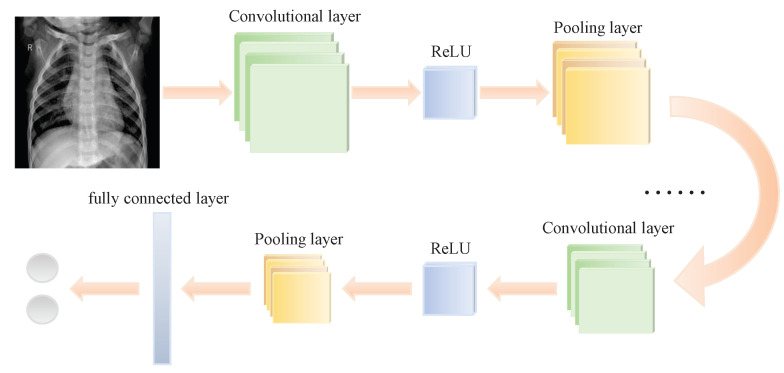
Basic structure of a convolutional neural network.

**Figure 2 entropy-24-01434-f002:**
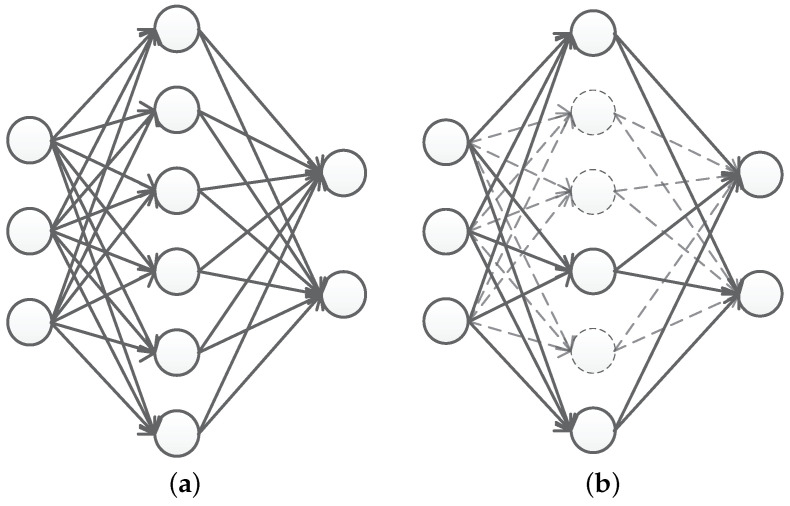
Standard neural network and neural network after application of dropout: (**a**) standard neural network; (**b**) neural network after application of dropout.

**Figure 3 entropy-24-01434-f003:**
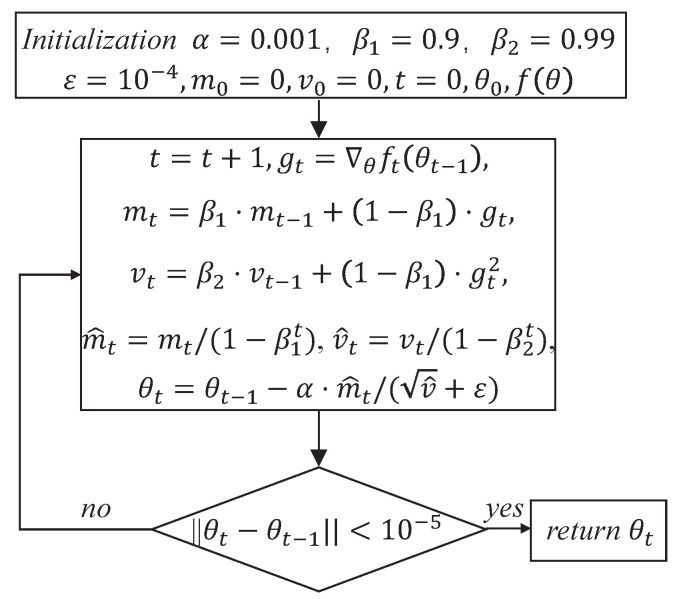
Adam algorithm flowchart.

**Figure 4 entropy-24-01434-f004:**
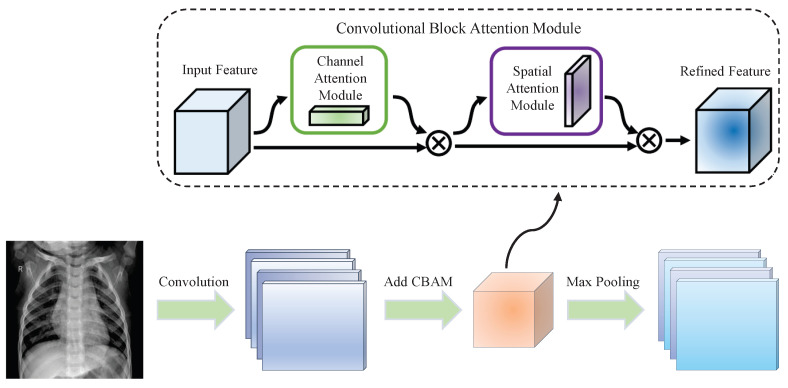
Convolutional neural network after adding the attention mechanism.

**Figure 5 entropy-24-01434-f005:**
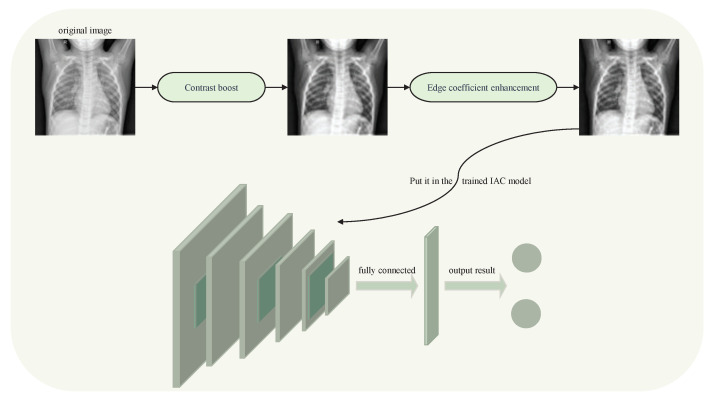
The overall framework of the CIEA method identifying athlete pneumonia.

**Figure 6 entropy-24-01434-f006:**
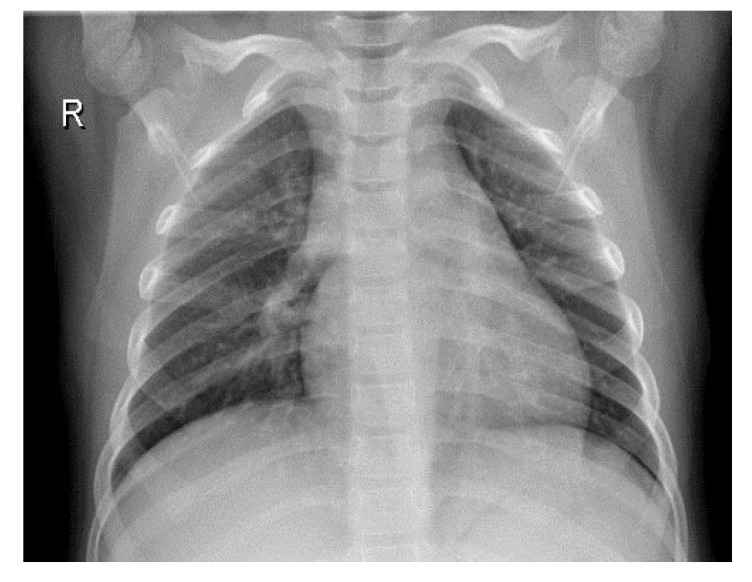
Lung image without pneumonia.

**Figure 7 entropy-24-01434-f007:**
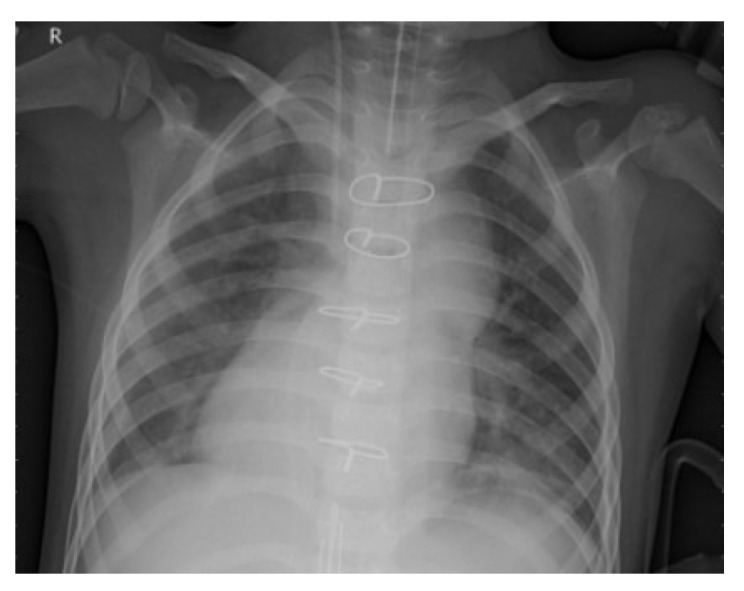
Lung image with pneumonia.

**Figure 8 entropy-24-01434-f008:**
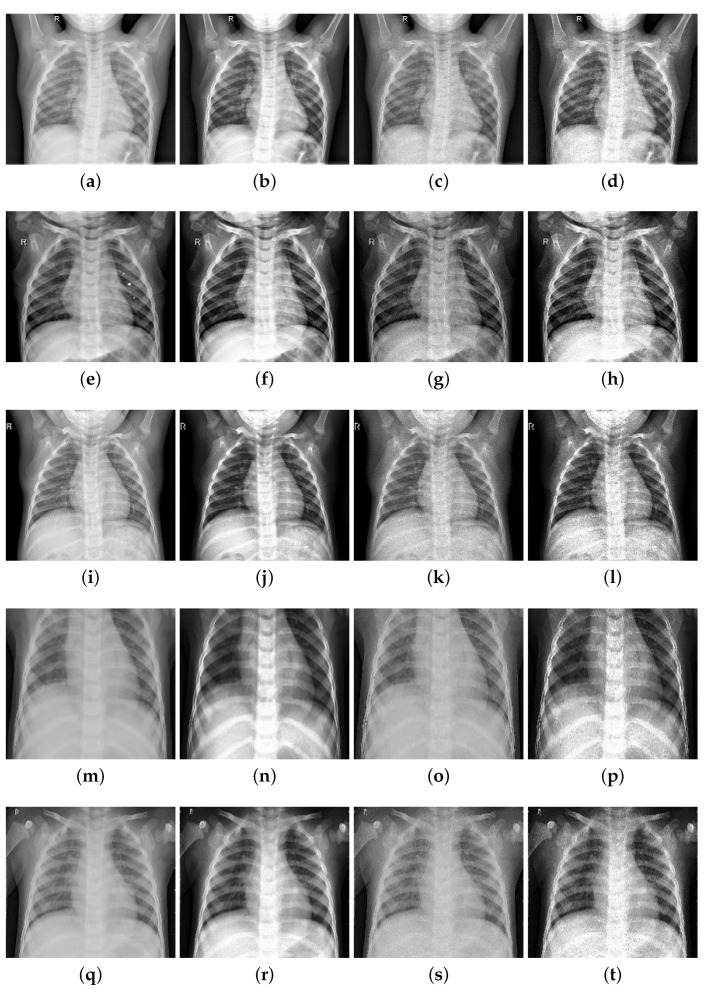
Image enhancement effect comparison image: (**a**) original-1; (**b**) contrast boost-1; (**c**) enhanced edge-1; (**d**) integrated-1; (**e**) original-2; (**f**) contrast boost-2; (**g**) enhanced edge-2; (**h**) integrated-2; (**i**) original-3; (**j**) contrast boost-3; (**k**) enhanced edge-3; (**l**) integrated-3; (**m**) original-4; (**n**) contrast boost-4; (**o**) enhanced edge-4; (**p**) integrated-4; (**q**) original-5; (**r**) contrast boost-5; (**s**) enhanced edge-5; (**t**) integrated-5.

**Figure 9 entropy-24-01434-f009:**
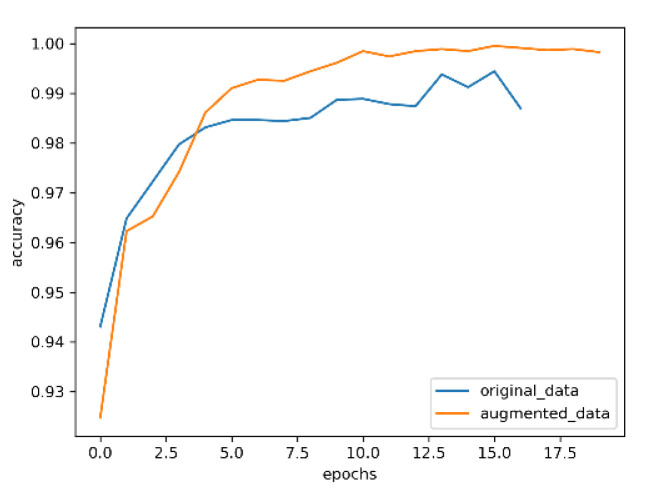
Accuracy of training set.

**Figure 10 entropy-24-01434-f010:**
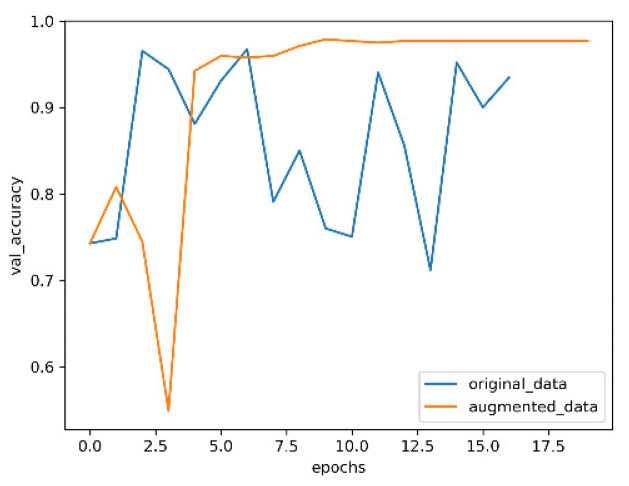
Accuracy of validation set.

**Figure 11 entropy-24-01434-f011:**
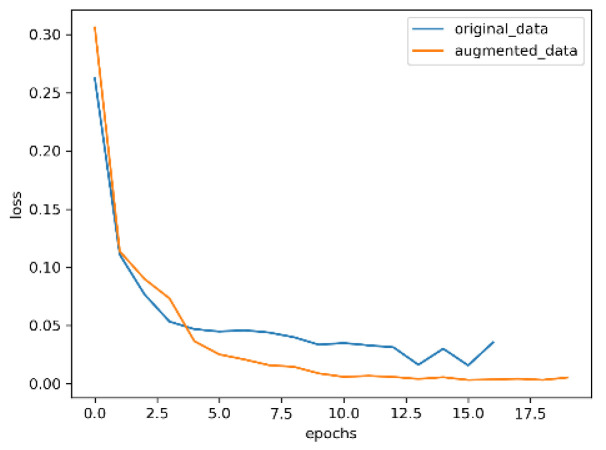
Training loss.

**Figure 12 entropy-24-01434-f012:**
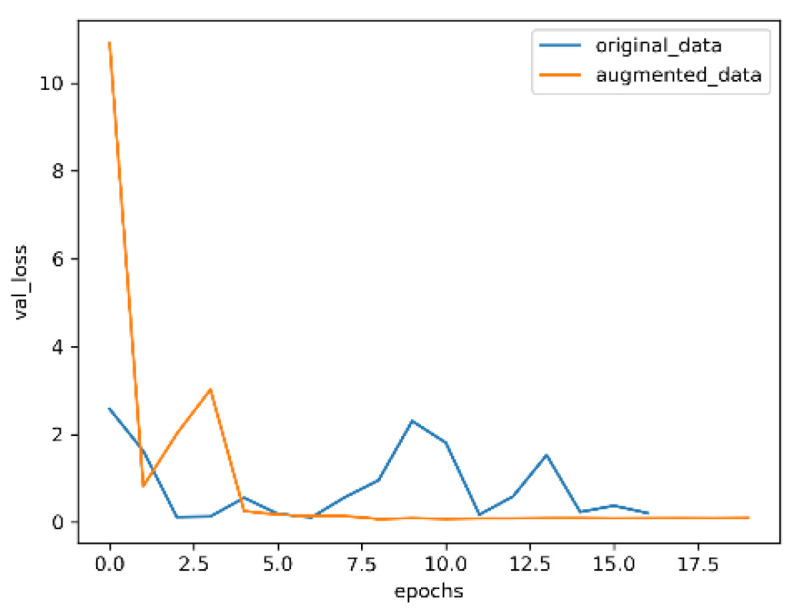
Validation loss.

**Table 1 entropy-24-01434-t001:** Accuracy of recognition training set, verification set and test set before and after enhancement.

Model Name	Training	Validation	Test
cnn_with_original_data	99.45%	93.28%	91.02
cnn_with_augmented_data	99.85%	97.31%	93.21%

**Table 2 entropy-24-01434-t002:** Accuracy for training set, verification set and test set classified by different algorithms.

Model Name	Train	Val	Test
DecisionTree	93.97%	87.55%	81.25%
RandomForest	99.72%	88.12%	81.25%
ChexNet	92.83%	83.32%	76.8%
CIEA	99.85%	97.31%	93.21%

## Data Availability

Not applicable.

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
