# Peer review of "Optimized Convolutional Neural Network Recognition for Athletes’ Pneumonia Image Based on Attention Mechanism"

_entropy, 2022, doi:10.3390/e24101434_

Round 1

Reviewer 1 Report (Previous Reviewer 2)

At first, I would like apologize for mistaking Akgundogdu Abdurrahim’s 5856 images with the 5612 images of the study. The authors have excellently used a better version of ReLU i.e. LeakyReLU and also had made relevant changes in the manuscript. But, still some changes cannot be ignored -

1.     Page 14, 6th line it will be “Random” instead of “Dandom”.

2.     Although the authors have clearly mentioned the sources of the X-ray images in the rebuttal, but please mention the sources of 5612 X-Ray images more elaborately in the manuscript or as a supplementary file.

3.     The authors have not yet defended that how their model is better than similar CNN models for detecting pneumonia chest X-rays which has similar accuracy. For example, https://doi.org/10.3390/app11031242. 

Author Response

Reviewer 2 Report (Previous Reviewer 4)

This is an interesting paper. Still the methodologies need to be tested in larger datasets. Two issues:

1) The authors use CNN, and a 3x3 kernel. They say they tested the kernel size. On the other hand usually the kernel size is related to the spectral components related to the sampling increment step so as to preserve the harmonics and information of the images.

2) Have the authors considered the use of grad-cam for better understanding and classification?

Author Response

This manuscript is a resubmission of an earlier submission. The following is a list of the peer review reports and author responses from that submission.

Round 1

Reviewer 1 Report

Comment 1: The authors have presented the application of the attention module (CBAM) for the classification of Lung x-ray images. However, they didn’t mention the architecture of CNN where they utilized the CBAM modules. Without backbone architecture, it’s complicated to judge the depth of CBAM applicability in the network.

Comment 2: Apart from CNN architecture, the authors didn’t compare CBAM application on other architecture and its feasibility with different architecture.

Comment 3: The loss function used in the experiment is missing. However, they mentioned ADAM optimizer for loss function optimization, but which loss function? Provide also the equation of the loss function.

Comment 4: Section 5.3.2.: “The algorithm first extracts features from the image using a two-dimensional discrete wavelet transform.”

The sentence should be: “The algorithm first extracts feature from the image using a two-dimensional discrete wavelet transform.”

Comment 5: Authors should compare the results with other standard deep learning CNN networks such as VGG16/19, Xception, ResNet, etc.

Comment 6: The benchmarking table comparing the proposed work with other existing state-of-the-art methods is missing. This table has 7-8 rows corresponding to further studies or publications and 7-8 columns corresponding to different demographics, AI architecture, performance, and clinical evaluation attributes.

Reviewer 2 Report

In this study, the authors generated a model for early detection of Pneumonia using chest X-Rays. Similar types of models where optimized convolutional neural networks with attention mechanism is used to identify the athletes’ lung images have been done earlier. Specially in this paper (https://doi.org/10.3390/app11031242) similar technique was used and similar accuracy was obtained. In the present study, the image processing by enhancing the images seems to automate image correction and thus only this part might be novel.  The authors should justify how their model is unique and how does it fill in the lacuna in existing models.

Also, these problems must be addressed -

1.Why in Algorithm 1 and 2 set step= 0.15 for [0,1] interval? Setting step = 0.1 would be perfect for matching [0,1] interval gap.

2. Algorithm 1 step 2 and 3 are repeated.

3. Abbreviation of ReLU is required when first used.

4. Although several steps for stopping the model from overfitting was adopted, How was the problem of dying ReLUs resolved for the model?

5. Correct recognition spelling in page 9

6. It will be “The algorithm” in page 13.

7. What is the source of the X-Ray images used in the study?

8. Why does the number of X- Ray images change from 5612 to 5856 while testing the model? Similar data can be used in both scenarios.

Reviewer 3 Report

In this work, the authors proposed a method for recognizing pneumonia from X-ray images, which consists of the following two stages:

1. An image enhancement method that boosts image contrast and highlights edges in the image by processing the coefficients in the curvelet domain. 

2. Then, a convolution neural network with attention modules is used to classify whether the patient has pneumonia or not.

Experiments were performed to compare the proposed method and other baselines, including decision tree and random forests. Results show that the proposed method achieves higher accuracy compared to the baselines. 

While this work is interesting, there are some concerns the authors should address prior to publication.

Major comments:

1. The authors added the attention module to the CNN. Ablation studies should be performed to verify that the attention module can improve the performance of the CNN.

2. Which dataset did the authors use to train the baseline models, original or enhanced? If the authors used the original data to train the baselines. For a fair comparison, baselines trained on the enhanced dataset should be also included in the experiments.

Reviewer 4 Report

This is an interesting study that uses DL for the detection of athlete's pneumonia. The methodology is well described with clear figures and appropriate detail. The authors should revise the paper in the following areas:

a) Expamd on the literature of AI/DL for pneumonia detection. Due to COVID, there is a large number of literature tackling pneumonia detection as well other lung functionality problems using X-Rays. Please identify the most important studies no matter if this work refers to athlete's pneumonia.

b) The dataset should be more clearly described. Both the subject's demographics, as welll as type of sport they participate is important since lung is not stressed in the same way, and this may have an effect in identifying pneumonia.

c) I would like to see a more wide discussion on the merits of the method comparing with other analogous methods. DL by itself is not innovative anymore.